# AdaTranS: Adapting with Boundary-based Shrinking for End-to-End Speech Translation

**Xingshan Zeng, Liangyou Li, Qun Liu**
Huawei Noah's Ark Lab
{zeng.xingshan,liliangyou,qun.liu}@huawei.com

## Abstract

To alleviate the data scarcity problem in End-to-end speech translation (ST), pre-training on data for speech recognition and machine translation is considered as an important technique. However, the modality gap between speech and text prevents the ST model from efficiently inheriting knowledge from the pre-trained models. In this work, we propose AdaTranS for end-to-end ST. It adapts the speech features with a new shrinking mechanism to mitigate the length mismatch between speech and text features by predicting word boundaries. Experiments on the MUST-C dataset demonstrate that AdaTranS achieves better performance than the other shrinking-based methods, with higher inference speed and lower memory usage. Further experiments also show that AdaTranS can be equipped with additional alignment losses to further improve performance.

## 1 Introduction

End-to-end speech translation (ST), which directly translates source speech into text in another language, has achieved remarkable progress in recent years (Berard et al., 2016; Duong et al., 2016; Weiss et al., 2017; Berard et al., 2018; Xu et al., 2021; Ye et al., 2022). Compared to the conventional cascaded systems (Ney, 1999; Mathias and Byrne, 2006), the end-to-end models are believed to have the advantages of low latency and less error propagation. A well-trained end-to-end model typically needs a large amount of training data. However, the available direct speech-translation corpora are very limited (Di Gangi et al., 2019). Given the fact that data used for automatic speech recognition (ASR) and machine translation (MT) are much richer, the paradigm of "pre-training on ASR and MT data and then fine-tuning on ST" becomes one of the approaches to alleviate the data scarcity problem (Bansal et al., 2019; Xu et al., 2021).

It has been shown that decoupling the ST encoder into acoustic and semantic encoders is beneficial to learn desired features (Liu et al., 2020; Zeng et al., 2021). Initializing the two encoders by pre-trained ASR and MT encoders, respectively, can significantly boost the performance (Xu et al., 2021). However, the modality gap between speech and text might prevent the ST models from effectively inheriting the pre-trained knowledge (Xu et al., 2021).

The modality gap between speech and text can be summarized as two dimensions. First, the length gap – the speech features are usually much longer than their corresponding texts (Chorowski et al., 2015; Liu et al., 2020). Second is the representation space gap. Directly fine-tuning MT parameters (semantic encoder and decoder) with speech features as inputs, which learned independently, would result in sub-optimal performance. Previous work has explored and proposed several alignment objectives to address the second gap, e.g., Cross-modal Adaption (Liu et al., 2020), Cross-Attentive Regularization (Tang et al., 2021a) and Cross-modal Contrastive Learning (Ye et al., 2022).

A shrinking mechanism is usually used to address the length gap. Some leverage Continuous Integrate-and-Fire (CIF) (Dong and Xu, 2020) to shrink the long speech features (Dong et al., 2022; Chang and Lee, 2022), but they mostly work on simultaneous ST and need extra efforts to perform better shrinking. Others mainly depend on the CTC greedy path (Liu et al., 2020; Gaido et al., 2021), which might introduce extra inference cost and lead to sub-optimal shrinking results. AdaTranS uses a new shrinking mechanism called boundary-based shrinking, which achieves higher performance.

Through extensive experiments on the MUST-C (Di Gangi et al., 2019) dataset, we show that AdaTranS is superior to other shrinking-based methods with a faster inference speed or lower memory usage. Further equipped with alignment objectives, AdaTranS shows competitive performance compared to the state-of-the-art models.

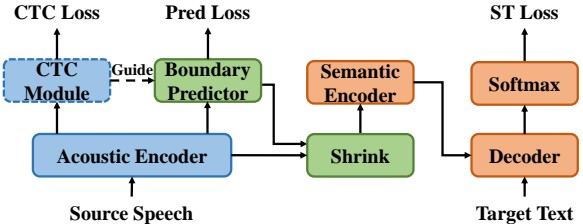

Figure 1: AdaTranS Architecture, where the blue modules are initialized with the ASR model and the orange modules are initialized with the MT model. The CTC module (dotted) can be removed during inference.

## 2 Proposed Model: AdaTranS

### 2.1 Problem Formulation

An ST corpus is denoted as $\mathcal{D}_{ST} = \{(\boldsymbol{x}, \boldsymbol{z}, \boldsymbol{y})\}$, containing triples of speech, transcription and translation. Here $\boldsymbol{x} = (x_1, x_2, ..., x_{T_x})$ is a sequence of speech features or waves as speech input, while $\boldsymbol{z} = (z_1, z_2, ..., z_{T_z})$ and $\boldsymbol{y} = (y_1, y_2, ..., y_{T_y})$ are the corresponding transcription in source language and translation in target language, respectively. $T_x$, $T_z$, and $T_y$ are the lengths of speech, transcription and translation, respectively, where usually $T_x \gg T_z$ and $T_x \gg T_y$.

### 2.2 Architecture

AdaTranS decouples the ST encoder into an acoustic encoder and a semantic encoder. To bridge the modality gap between speech and text, an adaptor is usually needed before the semantic encoder. We choose the shrinking operation (Liu et al., 2020; Zeng et al., 2021) as our adaptor, where the long speech sequences are shrunk to the similar lengths as the transcription based on designed mechanisms (details will be introduced in the next subsection). The shrunk representations are sent to the semantic encoder and ST decoder for output:

$$\mathcal{L}_{ST} = - \sum_{|\mathcal{D}_{ST}|} \sum_{t=1}^{T_y} \log p(y_t | y_{<t}, \boldsymbol{x}) \quad (1)$$

To incorporate extra ASR and MT data, we use the pre-trained ASR encoder to initialize the ST acoustic encoder, and the pre-trained MT encoder and decoder to initialize the ST semantic encoder and decoder, respectively. Both pre-trained models are first trained with extra ASR (or MT) data and then fine-tuned with the in-domain data (the ASR part or MT part in the ST corpus). Figure 1 displays our architecture as well as the training process.

### 2.3 Boundary-based Shrinking Mechanism

Previous shrinking mechanisms (Liu et al., 2020; Zeng et al., 2021) mostly depend on a CTC module (Graves et al., 2006) to produce token-label probabilities for each frame in the speech representations. Then, a word boundary is recognized if the labeled tokens of two consecutive frames are different. There are two main drawbacks to such CTC-based methods. First, the word boundaries are indirectly estimated and potentially affected by error propagation from the token label predictions which are usually greedily estimated by the argmax operation on the CTC output probabilities. Second, the token labels are from a large source vocabulary resulting in extra parameters and computation cost in the CTC module during inference.

We introduce a boundary-based shrinking mechanism to address the two drawbacks. A boundary predictor is used to directly predict the probability of each speech representation being a boundary, which is then used for weighted shrinking. Since the boundary labels on the speech representations are unknown during training, we introduce signals from the CTC module to guide the training of the boundary predictor. The CTC module will be discarded during inference. Below shows the details.

**CTC module.** We first briefly introduce the CTC module. It predicts a path $\boldsymbol{\pi} = (\pi_1, \pi_2, ..., \pi_{T_x})$, where $T_x$ is the length of hidden states after the acoustic encoder. And $\pi_t \in \mathcal{V} \cup \{\phi\}$ can be either a token in the source vocabulary $\mathcal{V}$ or the blank symbol $\phi$. By removing blank symbols and consecutively repeated labels, denoted as an operation $\mathcal{B}$, we can map the CTC path to the corresponding transcription. A CTC loss is defined as the probability of all possible paths that can be mapped to the ground-truth transcription $\boldsymbol{z}$:

$$\mathcal{L}_{CTC} = - \sum_{|\mathcal{D}_{ST}|} \sum_{\boldsymbol{\pi} \in \mathcal{B}^{-1}(\boldsymbol{z})} \log p(\boldsymbol{\pi} | \boldsymbol{x}) \quad (2)$$

**CTC-guided Boundary Predictor.** We propose to use a boundary predictor to replace the CTC module, which has a similar architecture but with only three labels. The three labels are <BK> (blank label), <BD> (boundary label) and <OT> (others), respectively. However, the ground-truth labels for training the predictor are unknown. Therefore, we introduce soft training signals based on the output probabilities of the CTC module. Specifically, the ground-truth probabilities of each frame $t$ to be

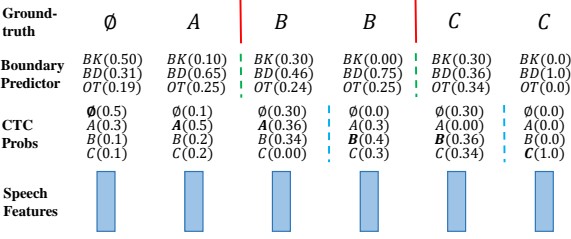

| Ground-truth | $\emptyset$ | $A$ | $B$ | $B$ | $C$ | $C$ |
|---|---|---|---|---|---|---|
| Boundary Predictor | BK(0.50) BD(0.31) OT(0.19) | BK(0.10) BD(0.65) OT(0.25) | BK(0.30) BD(0.46) OT(0.24) | BK(0.00) BD(0.75) OT(0.25) | BK(0.30) BD(0.36) OT(0.34) | BK(0.0) BD(1.0) OT(0.0) |
| CTC Probs | $\emptyset$(0.5) A(0.3) B(0.1) C(0.1) | $\emptyset$(0.1) A(0.5) B(0.2) C(0.2) | $\emptyset$(0.30) A(0.36) B(0.34) C(0.00) | $\emptyset$(0.0) A(0.36) B(0.4) C(0.3) | $\emptyset$(0.30) A(0.00) B(0.36) C(0.34) | $\emptyset$(0.0) A(0.0) B(0.0) C(1.0) |
| Speech Features | | | | | | |

Figure 2: An example of CTC probabilities (we assume there are three tokens in vocabulary for simplification), together with the corresponding boundary predictor soft signals and ground labels. We will get different boundary detection results based on their label probabilities (the solid and dotted lines). CTC argmax path predicts the wrong boundaries and we can get correct boundaries if we set the threshold $\theta$ as 0.5 in boundary predictor.

labeled as the three labels are defined as:

$$p'_t(\text{<BK>}) = p(\pi_t = \phi)$$
$$p'_t(\text{<BD>}) = \sum_{i \neq \phi} p(\pi_t = i)p(\pi_{t+1} \neq i) \quad (3)$$
$$p'_t(\text{<OT>}) = 1 - p'_t(\text{<BK>}) - p'_t(\text{<BD>})$$

Then, the objective for the boundary predictor[1] is:

$$\mathcal{L}_{Pred} = - \sum_{|\mathcal{D}_{ST}|} \sum_{t=1}^{T'_x} \sum_{i \in \Delta} p'_t(i) \log p_t(i) \quad (4)$$

where $\Delta = \{\text{<BK>}, \text{<BD>}, \text{<OT>}\}$. The CTC module is only used in the training process and can be discarded during inference. Since the number of labels in the predictor is significantly smaller than the size of the source vocabulary, the time and computation costs introduced by the predictor are negligible. Figure 2 shows an example to elaborate the advantage of such a predictor.

**Weighted Shrinking.** For shrinking, we define boundary frames as those with the probabilities of the <BD> label higher than a pre-defined threshold $\theta$. The frames between two boundary frames are defined as one segment, which can be aligned to one source token. Inspired by Zeng et al. (2021), we sum over the frames in one segment weighted by their probabilities of being blank labels to distinguish informative and non-informative frames:

$$\boldsymbol{h}_{t'}^A = \sum_{t \in seg\, t'} \boldsymbol{h}_t^A \frac{\exp(\mu(1 - p_t(\text{<BK>})))}{\sum_{s \in seg\, t'} \exp(\mu(1 - p_s(\text{<BK>})))} \quad (5)$$

where $\mu \geq 0$ denotes the temperature for the Softmax Function.

---

[1] Since the training of the predictor highly depends on the quality of the CTC output, the CTC module is also pre-trained.

**Forced Training.** We introduce a forced training trick to explicitly solve the length mismatch between speech and text representations. During training, we set the threshold $\theta$ dynamically based on the length of $\boldsymbol{z}$ to make sure the shrunk representations have exactly the same lengths as their corresponding transcriptions. Specifically, we first sort the probabilities to be <BD> of all frames in descending order, and then select the $T_z$-th one as the threshold $\theta$.

### 2.4 Training Objectives

The total loss of our AdaTranS will be:

$$\mathcal{L} = \mathcal{L}_{ST} + \alpha \cdot \mathcal{L}_{CTC} + \beta \cdot \mathcal{L}_{Pred} \quad (6)$$

where $\alpha$, $\beta$ are hyper parameters that control the effects of different losses.

## 3 Experiments

### 3.1 Experimental Setup

**Datasets.** We conduct experiments on three language pairs of MUST-C dataset (Di Gangi et al., 2019): English-German (En–De), English-French (En–Fr) and English-Russian (En–Ru). We use the official data splits for train and development and tst-COMMON for test. We use LibriSpeech (Panayotov et al., 2015) to pre-train the acoustic model. OpenSubtitles2018[2] or WMT14[3] are used to pre-train the MT model. The data statistics are listed in Table 3 of Appendix A.

**Preprocessing.** We use 80D log-mel filterbanks as speech input features and SentencePiece[4] (Kudo and Richardson, 2018) to generate subword vocabulary with a size of 16000 for each language pair. More details please refer to the Appendix B.

**Model Setting.** Conv-Transformer (Huang et al., 2020) or Conformer (Gulati et al., 2020) (results in Table 2 are achieved by AdaTranS with Conformer, while the remaining results utilized Conv-Transformer) is used as our acoustic encoder, both containing 12 layers. For the semantic encoder and ST decoder, we follow the general NMT Transformer settings (i.e., both contain 6 layers). Each Transformer layer has an input embedding dimension of 512 and a feed-forward layer dimension of 2048. The whole model contains 107M (140M if

---

[2] http://opus.nlpl.eu/OpenSubtitles-v2018.php
[3] https://www.statmt.org/wmt14/translation-task.html
[4] https://github.com/google/sentencepiece

| Model | Diff≤2 (%) | Speedup | Mem Usage | BLEU En-De | En-Fr |
|---|---|---|---|---|---|
| No Shrink | – | 1.00× | 1.00 | 26.0 | 36.8 |
| Fix Shrink | 36.7 | **1.06×** | **0.74** | 25.4 | 36.0 |
| CIF-Based | 70.3 | 1.04× | **0.74** | 25.8 | 36.2 |
| CTC-Based | 80.2 | 0.76× | 1.77 | 26.4 | 36.9 |
| Boundary-Based | | | | **26.7** | **37.4** |
| - Forced Train | **81.9** | **1.06×** | 0.78 | 26.4 | 37.1 |
| - Blank Label | | | | 26.3 | 36.6 |

Table 1: The results of shrinking-based methods and the corresponding shrinking quality, evaluated with length differences between the shrunk representations and transcriptions. Diff≤2 means the length differences are less than or equal to 2. The speedup and memory usage are both tested with a batch size of 16, and we only display the relative values for clear comparison.

| Model | BLEU En-De | En-Fr | En-Ru |
|---|---|---|---|
| MT | 34.4 | 44.9 | 21.3 |
| Cascaded Model | 28.1 | 37.0 | 17.6 |
| JT-S-MT (Tang et al., 2021a) | 26.8 | 37.4 | – |
| Chimera (Han et al., 2021) | 27.1 | 35.6 | 17.4 |
| XSTNet (Ye et al., 2021) | 27.1 | 38.0 | 18.5 |
| SATE (Xu et al., 2021) | 28.1 | – | – |
| STEMM (Fang et al., 2022) | **28.7** | 37.4 | 17.8 |
| ConST (Ye et al., 2022) | 28.3 | 38.3 | 18.9 |
| AdaTranS | **28.7** | **39.2** | **19.3** |
| - Boundary-based Shrink | 28.1 | 38.8 | 18.8 |

Table 2: Comparisons with the SOTA models. The first two rows are the results with our pre-trained MT model and cascading pre-trained ASR and MT models.

use Conformer) parameters. The hyper-parameters in Eq. 6 are set as: $\alpha = 1.0$ and $\beta = 1.0$, respectively. The temperature of the softmax function in Eq. 5 ($\mu$) is 1.0, while the threshold $\theta$ in the boundary predictor is set to 0.4 during inference[5]. Training details please refer to Appendix B.

We apply SacreBLEU[6] for evaluation, where case-sensitive detokenized BLEU is reported.

### 3.2 Experiment results

Table 1 compares different shrinking-based methods in terms of quality and efficiency (for fair comparison, we use the same architecture pre-trained with the same data and add a CTC loss to all the compared models). Besides translation quality, we use length differences between the shrunk representations and the corresponding transcriptions to evaluate shrinking quality following Zeng et al. (2021). We use inference speedup and memory usage to evaluate efficiency.

For comparisons, the Fix-Shrink method shrinks the speech features with a fixed rate (e.g. every 3 frames). The CIF-Based method (Dong et al., 2022) is based on a continuous integrate-and-fire mechanism. The CTC-Based method (Liu et al., 2020) shrinks features based on CTC greedy paths. As can be seen, poor shrinking (Fix-Shrink and CIF-based) hurts the performance, although with better efficiency. The boundary-based shrinking used in AdaTranS and the CTC-based method achieve better shrinking quality, with performance improved. However, CTC-Based method hurts the inference

efficiency (lower inference speed and higher memory usage) as they introduce extra computation cost producing greedy CTC path in a large source vocabulary. Our method performs the best in both shrinking and translation quality with nice inference efficiency. On the other hand, we also notice that removing forced training trick ("-Forced Train") or weighted-shrinking (i.e., "-Blank Label", simply average the frame representations rather than use Eq. 5) will affects the translation quality, showing the effectiveness of these two components.

**Adopting Alignment Objectives.** AdaTranS can be further improved with objectives that align speech and text representations (i.e. bridging the representation space gap introduced in Section 1). Table 2 shows the results of AdaTranS equipped with Cross-modal Contrastive (Ye et al., 2022) and knowledge distillation (KD) guided by MT, compared to the models that also work on modality alignment objectives (more complete comparisons please refer to Appendix C). The results show that AdaTranS achieves competitive results in all three datasets. We also examine and show the effects of boundary-based shrinking in such setting ("-Boundary-base Shrink").

**Influence of the Threshold.** We also examine the threshold $\theta$ for the boundary predictor. Figure 3 shows the distribution of the predicted boundary probability (i.e. $p_t(\texttt{<BD>})$) for each frame. We find that the boundary predictor is confident ($< 0.1$ and $> 0.9$) in most cases. However, even though only a small portion of predictions are in the range of $[0.1, 0.9]$, they significantly affect the BLEU scores when the threshold changes (the red line in Figure 3). The model achieves the best perfor-

---

[5]All the hyper-parameters are set through grid search based on the performance of the development set.

[6]https://github.com/mjpost/sacreBLEU

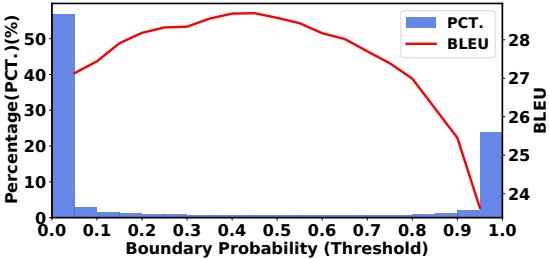

Figure 3: The distribution of probabilities for boundary prediction, and the corresponding BLEU scores using different values as threshold in MUST-C En-De test set.

mance when the threshold is around $0.4$.

## 4 Related Work

Numerous techniques have been proposed to adapt speech and text representations in order to mitigate the modality gap in end-to-end ST. Wang et al. (2020) introduce noise into the text input, while Salesky and Black (2020); Tang et al. (2021a,b) employ phoneme sequences for text input. These approaches reduce differences between speech and text input by extending text representations. Conversely, some research aims at shrinking lengthy speech input by presenting various shrinking strategies (Salesky et al., 2019; Dong et al., 2020, 2021; Liu et al., 2020; Gaido et al., 2021; Zeng et al., 2021). Our work falls within this category, proposing a new approach that offers enhanced effectiveness and efficiency. Others prioritize aligning the speech-text representation space (Liu et al., 2020; Tang et al., 2021a; Xu et al., 2021; Ye et al., 2022; Zhang et al., 2022; Le et al., 2023).

CTC alignment's role in enhancing ST is particularly pertinent to our research. Earlier studies have demonstrated that integrating an additional CTC loss for multi-task learning or pre-training assists in ST model training (Wang et al., 2020; Xu et al., 2021). Some also leverage CTC alignment to shrink speech inputs (Salesky et al., 2019; Liu et al., 2020; Gaido et al., 2021; Zeng et al., 2021). Moreover, Le et al. (2023) and Zhang et al. (2023) delve deeper, optimizing CTC objectives for ST.

## 5 Conclusion

This work proposes a new end-to-end ST model called AdaTranS, which uses a boundary predictor trained by signals from CTC output probabilities, to adapt and bridge the length gap between speech and text. Experiments show that AdaTranS per-

forms better than other shrinking-based methods, in terms of both quality and efficiency. It can also be further enhanced by modality alignment objectives to achieve state-of-the-art results.

## Limitations

**Learning of the proposed boundary predictor.** The learning objective of our boundary predictor is constructed by the soft labels from CTC objective. Since the labels are not accurate labels, it is inevitable to introduce errors during training. However, the groundtruth labels for boundary predictor is difficult to obtain. One alternative is to use forced alignment tools. That also introduces other problems. First, off-the-shelf forced alignment tools only support speech in popular languages, which limits the use of the method to other languages. Second, forced alignment also doesn't guarantee the correctness of labeling, and we still need to further approximate the labeling results when applying them to speech features after acoustic encoder (with 4x or 8x downsampling).

**Sensitivity of the selective threshold.** From Figure 3, we can find that the BLEU score is sensitive to the threshold selection, although the boundary predictor is confident in most cases. We leave it to our future work to alleviate this phenomenon.

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

## A  Data Statistics.

Table 3 shows the data statistics of the used datasets. ST datasets are all from MUST-C, and LibriSpeech serves as extra ASR data. MT data either comes from OpenSubtitles2018 or WMT14 following settings of previous work.

| Corpus | ST(Hours/#Sents) | ASR(Hours) | MT(#Sents) |
|--------|------------------|------------|------------|
| En–De  | 408/234K         | 960        | 18M(OS)    |
| En–Fr  | 492/280K         | 960        | 18M(WMT)   |
| En–Ru  | 489/270K         | 960        | 2.5M(WMT)  |

Table 3: The statistics for the three language pairs. OS: OpenSubtiles2018. WMT: WMT14.

## B  Implementation Details.

**Data Preprocessing.**  We use 80D log-mel filterbanks as speech input features, which are calculated with 25ms window size and 10ms step size and normalized by utterance-level Cepstral Mean and Variance Normalization (CMVN). All the texts in ST and MT data are preprocessed in the same way, which are case-sensitive with punctuation preserved. For training data, we filter out samples with more than 3000 frames, over 256 tokens, or whose ratios of source and target text lengths are outside the range [2/3, 3/2]. We use SentencePiece (Kudo and Richardson, 2018) to generate subword vocabulary for each language pair. Each vocabulary is learned on all the texts from ST and MT data and shared across source and target languages, with a total size of 16000.

**Training Details.**  We train all the models using Adam optimizer (Kingma and Ba, 2015) with a 0.002 learning rate and 10000 warm-up steps followed by the inverse square root scheduler. Label smoothing and dropout strategies are used, both set to 0.1. The models are fine-tuned on 8 NVIDIA Tesla V100 GPUs with 40000 steps, which takes about 10 hours in average. The batch size is set to 40000 frames per GPU. We save checkpoints every epoch and average the last 10 checkpoints for evaluation with a beam size of 10.

## C  More Analysis.

**Complete Comparisons with the SOTA models.** Table 4 extends the results in Table 2 to include the detailed settings of the compared models, including external data used and training objectives.

We also include two SOTA works based on speech-text joint pre-training (ST Joint PT), which shows great improvements by applying complex joint pre-training objectives. It should be noted that it is not in line with the focus of this work and our Ada-TranS might also benefit from them by initializing the modules after such joint pre-training.

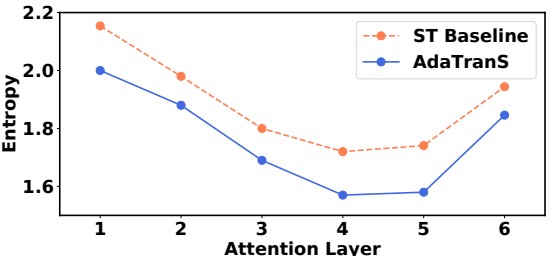

Figure 4: The attention entropy of each attention layer for end-to-end ST baseline and our model.

**Better Source-Target Alignment.**  We evaluate the entropy of the cross attention from the ST baseline (i.e. no any shrinking) and AdaTranS[7]. Let $\alpha_{ij}$ be the attention weight for a target token $y_i$ and a source speech feature (after shrinking) $x_j$, the entropy for each target token is defined as $E_i = -\sum_{j=1}^{|x|} \alpha_{ij} \log \alpha_{ij}$. We then average the attention entropy of all target tokens in the test set. Lower entropy means the attention mechanism is more confident and concentrates on the source-target alignment. Figure 4 shows the entropy of different decoder layers. AdaTranS exhibits consistently lower entropy than the ST baseline. This means that our shrinking mechanism improves the learning of attention distributions.

**Influence of Text Input Representations.**  Representing text input with phonemes helps reduce the differences between speech and text (Tang et al., 2021a,b). However, word representations and punctuation are important for learning semantic information, which are usually ignored when phonemes are used in prior works. Table 5 shows the MT results when using different text input representations, together with the ST performance that is initialized from the corresponding MT model. We can observe that the performance of downstream ST model is affected by the pre-trained MT model. Therefore, instead of following prior phoneme-level work for

---

[7]To fairly compare, we also shrink the speech features of the ST baseline with the same boundaries detected by our boundary predictor.

| Model | External Data | | | Training Detail | | | BLEU | | |
|---|---|---|---|---|---|---|---|---|---|
| | Speech | ASR | MT | ST Joint PT | Enc Align | KD | En-De | En-Fr | En-Ru |
| MT | – | – | – | – | – | – | 34.4 | 44.9 | 21.3 |
| Cascaded Model | – | – | – | – | – | – | 28.1 | 37.0 | 17.6 |
| STPT† (Tang et al., 2022) | ✓ | ✓ | ✓ | ✓ | ✗ | ✗ | 29.2 | 39.7 | – |
| SpeechUT† (Zhang et al., 2022) | ✓ | ✓ | ✓ | ✓ | ✗ | ✗ | 30.1 | 41.4 | – |
| JT-S-MT (Tang et al., 2021a) | ✗ | ✓ | ✓ | ✗ | ✓ | ✓ | 26.8 | 37.4 | – |
| Chimera (Han et al., 2021) | ✓ | ✗ | ✓ | ✗ | ✓ | ✗ | 27.1 | 35.6 | 17.4 |
| XSTNet (Ye et al., 2021) | ✓ | ✗ | ✓ | ✗ | ✗ | ✗ | 27.1 | 38.0 | 18.5 |
| SATE (Xu et al., 2021) | ✗ | ✓ | ✓ | ✗ | ✓ | ✓ | 28.1 | – | – |
| STEMM (Fang et al., 2022) | ✓ | ✗ | ✓ | ✗ | ✓ | ✗ | **28.7** | 37.4 | 17.8 |
| ConST (Ye et al., 2022) | ✓ | ✗ | ✓ | ✗ | ✓ | ✗ | 28.3 | 38.3 | 18.9 |
| AdaTranS | ✗ | ✓ | ✓ | ✗ | ✓ | ✓ | **28.7** | **39.2** | **19.3** |
| - Boundary-based Shrink | ✗ | ✓ | ✓ | ✗ | ✓ | ✓ | 28.1 | 38.8 | 18.8 |

Table 4: The complete comparisons with the SOTA models, including external data used and training objectives. "ST Joint PT" indicates speech-text joint pre-training. "Enc Align" means whether adding objectives to aligning representations of speech and text in encoder. "KD" means whether using knowledge distillation from MT.

| | SPM | SPM w/o Punct | Phoneme |
|---|---|---|---|
| **MT** | | | |
| 1. Only MUST-C Data | 30.7 | 28.3 | 28.2 |
| 2. PT with Extra Data | 34.4 | 31.6 | 29.5 |
| **ST** | | | |
| Initialized with Model 2 | 26.0 | 25.8 | 25.6 |

Table 5: BLEU scores of different text input representations in MUST-C En-De. SPM means using the subword units learned from a sentencepiece model, while "w/o Punct" indicates that punctuation is removed.

pre-training the MT, in this work we use subword units with punctuation and incorporate the shrinking mechanism to mitigate the length gap.

| Data | Extra MT Corpus | With MTL | W/O MTL |
|---|---|---|---|
| En-De | OS (18M) | 28.7 | 28.3 |
| En-Fr | WMT14 (18M) | 38.7 | 39.2 |
| En-Ru | WMT14 (2.5M) | 19.0 | 19.3 |

Table 6: BLEU scores of AdaTranS with KD using MT multi-task learning or not in different language pairs.

**Influence of Multi-task Learning (MTL).** Previous work usually applies multi-task learning together with MT task (adding external MT data) to improve performance. However, our experiments (Table 6) show that when using KD, MTL might not be always helpful, especially with MT corpus in different domains (OpenSutitles is in spoken language domain like ST, but data from WMT14 is mostly in news domain).