# OpenReview forum: "AdaTranS: Adapting with Boundary-based Shrinking for End-to-End Speech Translation"
_EMNLP/2023/Conference — EMNLP 2023 Findings_

### Official Review · Reviewer_kuHR · 2023-08-04

**Soundness:** 2

**Excitement:**

3: Ambivalent: It has merits (e.g., it reports state-of-the-art results, the idea is nice), but there are key weaknesses (e.g., it describes incremental work), and it can significantly benefit from another round of revision. However, I won't object to accepting it if my co-reviewers champion it.

**Paper Topic And Main Contributions:**

This paper proposes a mechanism for detecting word boundaries based on connectionist temporal classification (CTC) predictions. These predictions are useful to reduce the feature length mismatch for audio-text conversion pipelines, applied by the authors to the problem of (end to end) speech-to-text translation. The proposed method operates on each individual CTC prediction, obtaining a probability of

**Questions For The Authors:**

A) To what extent would the use of the pre-trained forced aligners to better train the predictor would help? If we ignore the potential limited availability in some languages of forced aligners (which do not require high amounts of data to train in general), it would provide a better understanding of what is the true potential of the proposed idea.

**Reasons To Accept:**

The proposed method is a simple-yet-effective mechanism for detecting boundaries that has a positive impact on the downstream task. Even if the proposal and results are on a still preliminary stage of exploration, the model could spark conversation in the field.

**Reasons To Reject:**

The authors highlight clearly the limitations I saw while reading the paper.

Most notably, I find that the proposed boundary prediction model discards very important information coming from the sequentiality of the predicted information, similarly to how common forced alignment modules commonly used to align speech and audio are implemented. As such, and without having an understanding of what is the actual 'accuracy' of the soft labels predicted by the module, it is hard to understand if the improvements the authors achieve are really because of the proposed algorithm.

**Reproducibility:**

4: Could mostly reproduce the results, but there may be some variation because of sample variance or minor variations in their interpretation of the protocol or method.

**Reviewer Confidence:**

3: Pretty sure, but there's a chance I missed something. Although I have a good feel for this area in general, I did not carefully check the paper's details, e.g., the math, experimental design, or novelty.

---

> ### Author Rebuttal · Authors · 2023-08-29
>
> Thanks for your constructed reviews! We truly appreciate the time and effort you invested in understanding our work!
>
> Regarding your concern about "the proposed boundary prediction model discards very important information coming from the sequentiality of the predicted information", we believe there might be some misunderstandings about our method:
>
> 1. Our method does not discard any information concerning sequentiality. The boundary predictor is based on the output from the acoustic encoder, either the Conv-Transformer or Conformer (as referenced in "Model setting" in Section 3.1). Both of these incorporate position information. Moreover, our prediction loss, as described in Eq. 4, is guided by the CTC loss, which is also computed considering the sequential target from the corresponding transcript.
>
> 2. We have not omitted any frame-level representations during our modeling. As indicated in Eq. 5, every representation is integrated into our weighted average process to generate the final acoustic features.
>
> In response to your another concern about "As such, and without having an understanding of what is the actual 'accuracy' of the soft labels predicted by the module, it is hard to understand if the improvements the authors achieve are really because of the proposed algorithm.", we'd like to clarify the following:
>
> 1. While the precise 'accuracy' of the soft labels is challenging to obtain — and indeed might be a subject for future research — it doesn't render our approximations void. Using the CTC objective for speech labeling is standard in speech recognition. Relying on its implicit guidance for boundary prediction is a natural extension. One contribution of our paper is experimentally proving the efficacy of such an approach.
>
> 2. The results in Table 1 lucidly depict the improvements facilitated by our method. To be explicit, we re-implemented all presented methods under identical settings. The sole differentiation lies in their method of shrinking acoustic features. A direct comparison between our "Boundary-Based" method and "No Shrink" reveals improvements in both translation quality and efficiency, thereby underscoring the potency of our proposed technique.
>
> In response to your inquiry, "To what extent would the use of the pre-trained forced aligners to better train the predictor help?":
>
> We did try a similar idea during our preliminary experiments. Specifically, we employed a pre-trained aligner sourced from Montreal Forced Aligner (https://github.com/MontrealCorpusTools/Montreal-Forced-Aligner). With this tool, we extracted alignments between speech and transcriptions to guide the shrinking of speech features. The outcome was impressive in terms of shrinking quality, achieving over 99% as per the "Diff≤2" metric in Table 1. However, the translation quality did not witness any significant leap: it was 26.5 BLEU for MUSTC En-De, marginally lower than the 26.7 BLEU achieved by our current method.

---

### Official Review · Reviewer_FEtD · 2023-08-05

**Soundness:** 4

**Excitement:**

4: Strong: This paper deepens the understanding of some phenomenon or lowers the barriers to an existing research direction.

**Paper Topic And Main Contributions:**

In this paper, AdaTranS is introduced as a novel method capable of directly converting spoken input from one language into written text of another language, all in a end-to-end manner.
A novel shrinking mechanism is proposed to address the length mismatch between speech and text features.
This involves employing a boundary predictor to assign probabilities to each speech representation, indicating the likelihood of it being a boundary. Subsequently, these probabilities are harnessed for a weighted shrinking process.

**Questions For The Authors:**

- How accurate is the boundary prediction? Is there an alternative method to assess its accuracy apart from relying on Figure 3?

**Reasons To Accept:**

- Unlike earlier methods that indirectly predict boundaries by relying on token outputs from CTC, the AdaTranS architecture employs a direct approach. It predicts the likelihood of word boundaries, mitigating the potential error propagation inherent in such indirect methods.
- The conducted experiments on the MUST-C dataset reveal that the introduced AdaTranS offers superior performance compared to alternative compression techniques. Notably, AdaTranS achieves faster inference for speech processing while maintaining low memory utilization.
- The evaluation also encompasses a cascaded approach, highlighting the superiority of the proposed end-to-end method.

**Reasons To Reject:**

- The authors assert their decision to not employ a forced alignment tool stems from its support being limited to widely spoken languages. Nevertheless, it remains unclear why training a forced alignment tool couldn't be pursued when AdaTranS has access to sufficient training data.

**Reproducibility:**

4: Could mostly reproduce the results, but there may be some variation because of sample variance or minor variations in their interpretation of the protocol or method.

**Reviewer Confidence:**

4: Quite sure. I tried to check the important points carefully. It's unlikely, though conceivable, that I missed something that should affect my ratings.

---

> ### Author Rebuttal · Authors · 2023-08-29
>
> Thank you for your constructive reviews! We truly appreciate the time and effort you've invested in understanding our work.
>
> Regarding your concern about why we didn't train a forced alignment tool when AdaTranS has access to sufficient training data: we actually conducted a similar preliminary experiment. Specifically, we used a pre-trained aligner from the Montreal Forced Aligner (https://github.com/MontrealCorpusTools/Montreal-Forced-Aligner). With this tool, we extracted alignments between speech and transcriptions to guide the reduction of speech features. The results were impressive in terms of the shrinking quality, achieving over 99% as per the "Diff≤2" metric shown in Table 1. However, the translation quality did not see any significant improvement: it was 26.5 BLEU for MUSTC En-De, marginally lower than the 26.7 BLEU achieved by our current method.
>
> In response to your question, "How accurate is the boundary prediction? Is there an alternative method to assess its accuracy apart from relying on Figure 3?":
>
> This remains an open question. As discussed in Section 2.2 and in the Limitations section, there's no precise way to obtain ground-truth labels for each frame, especially after downsampling. Therefore, we can only indirectly assess the "accuracy." Besides evaluating the final quality of downstream tasks (e.g., translation performance, as illustrated in Figure 3), alternative methods for assessment could include:
>
> 1. Evaluating the shrinking quality, which is measured by the length differences between the reduced representations and transcriptions (i.e., results shown in Table 1). Generally, better boundary prediction can result in better length alignment.
>
> 2. Using pseudo labels. We can pre-label each frame with pretrained speech labelers like CTC modules or forced aligners. However, this method might introduce noises, as no ground-truth labels are available.

---

### Official Review · Reviewer_g78k · 2023-08-08

**Soundness:** 4

**Excitement:**

3: Ambivalent: It has merits (e.g., it reports state-of-the-art results, the idea is nice), but there are key weaknesses (e.g., it describes incremental work), and it can significantly benefit from another round of revision. However, I won't object to accepting it if my co-reviewers champion it.

**Missing References:**

In addition to the important references cited in "Reasons to reject", I think the literature review could be greatly improved (without adding too much space) by briefly discussing the following references:
- L24: Berard et al., 2016 ("Listen and translate: A proof of concept for end-to-end speech-to-text translation") were among the first to work on end-to-end ST (concurrent with Duong et al., 2016 but Duong et al. focused on the alignment and reranking tasks instead of translation).
- Figure 1: Is the CTC module pre-trained as well? If so, Wang et al., 2020 ("Bridging the Gap between Pre-Training and Fine-Tuning for End-to-End Speech Translation") were the first to use ASR pre-training with CTC with the motivation to reduce subnet wastes while Le et al., 2023 ("Pre-training for Speech Translation: CTC Meets Optimal Transport") did a comprehensive study on the effect of ASR pre-training CTC.
- L226: Papineni et al., 2002 ("Bleu: a method for automatic evaluation of machine transla- tion"); Post, 2018 ("A call for clarity in reporting BLEU scores") proposed BLEU and sacreBLEU as translation metrics.
- L190: This work belongs to the multi-task learning paradigm in which ST is jointly trained with ASR (and maybe other losses). The landmarks papers in this direction are Anastasopoulos and Chiang, 2018 ("Tied multitask learning for neural speech translation") and Sperber et al., 2019 ("Atention-passing models for robust and data-efficient end-to-end speech translation").

**Paper Topic And Main Contributions:**

This paper proposes a novel boundary predictor mechanism to tackle the speech-text modality gap in end-to-end speech translation (ST). This boundary-based shrinking method consists of two parts. First, a boundary predictor module predicts the word boundaries based on the signals from the CTC module (by collapsing the vocabulary into three labels including blank, boundary, and others). Second, a shrinking operation is applied on top of the boundary prediction results to shrink the speech representations so as to mitigate the length mismatch between speech and text sequences. The proposed method is complementary with additional alignment losses.

**Questions For The Authors:**

- Is previous work comparable with this work in terms of ASR and MT pre-training? In particular, this work pre-trains ASR and MT models on external ASR and MT data, then fine-tunes these models on ASR and MT data of MuST-C. Are all methods in Table 1 pre-trained in the same way?
- Preprocessing: why is the choice of vocabulary size 16000?
- Table 1: What are the model architectures used in these methods? Why used Diff<2 instead of the percentage of lengths between shrunk speech and text?
- Do you have an intuition why weighted shrinking has a much bigger impact on En-Fr while it has only a marginal effect on En-De?
- What are the differences in AdaTranS between Table 1 and Table 2 that leads to the gain of 2 BLEU points in Table 2? Is it only thanks to the addition of cross-modal contrastive learning and KD?
- The label <BD> corresponds to one or multiple consecutive frames? What is the shrinking method in the latter case?
- Did you try using simpler shrinking operations such as averaging instead of weighted shrinking after the boundary predictor module?

**Reasons To Accept:**

- The paper is well-written and easy to follow. It is well-motivated and addresses an important problem in ST which is the speech-text modality gap.
- The proposed method is effective: yielding better performance than several existing shrinking-based methods being compared in the paper with faster inference speed and memory usage.
- Competitive performance with current strong ST systems using self-supervised pre-training methods.

**Reasons To Reject:**

1. This paper misses important references to highly related work and thus fails to properly position itself with respect to the existing literature:
	- Zhang et al., 2023 ("Efficient CTC Regularization via Coarse Labels for End-to-End Speech Translation") showed that simply coarse labeling CTC using simple heuristics rules such as truncation, division or modulo operations yields comparable-to-better performance than baselines with considerable training speedup.
	- Shrinking speech features has been an active research area and not limited to only CIF-based or CTC-based approach. Other methods including compressed phoneme-like features (Salesky et al., 2019. Exploring Phoneme-Level Speech Representations for End-to-End Speech Translation), adaptive feature selection (Zhang et al., 2020. Adaptive Feature Selection for End-to-End Speech Translation), and phoneme-as-MT-input (Salesky et al., 2020. Phone Features Improve Speech Translation and Tang et al., 2021. Improving Speech Translation by Understanding and Learning from the Auxiliary Text Translation Task) should be mentioned as well. In terms of CTC-based shrinking approach, an important and highly related work using very simple heuristics rules (Zhang et al., 2023. Efficient CTC Regularization via Coarse Labels for End-to-End Speech Translation) should be cited and included for comparison with the proposed method. This is alternative to the CTC greedy path approach that the authors mentioned.
	- Previous work using CTC alignments to improve ST is also highly relevant to the current paper (which also relies on CTC alignments signals) and therefore should be discussed. For example, Yan et al., 2022 ("CTC Alignments Improve Autoregressive Translation") and Le et al., 2023 ("Pre-training for Speech Translation: CTC Meets Optimal Transport"). In particular, Le et al. 2023 specifically address the speech-text modality gap using CTC (which is the main subject of the current submission), and also propose/discuss different length-matching mechanisms when comparing speech and text sequences.

2. The experimental settings are not clearly described in some parts.
	- The ablation study setting to compare different shrinking method is not clear: which method adopts Conv-Transformer and which uses Conformer (Section 3.1., "Model setting" only specifies that AdaTranS uses Conformer in Table 2, but what about Table 1?) Do all methods use the same model architecture in each experimental setting?
	- The way to incorporate additional alignment objectives is not presented. For example, it is not straightforward to me how KD guided by MT is added to the architecture in Figure 1. Did you perform KD with a frozen pre-trained MT model or did you add an additional MT loss and perform KD with the MT model being updated during training?

3. The comparison to SoTA methods has some issues. The caption of Table 2 is "Comparisons with the SOTA models..." but in reality Table 2 only contains "modality alignment" methods as stated in the text. Note that even within this "modality alignment" subgroup, the authors have missed some recent work such as Le et al. 2023 (Pre-training for Speech Translation: CTC Meets Optimal Transport) and Ouyang et al. 2023 (WACO: Word-Aligned Contrastive Learning for Speech Translation). I would suggest to fix the caption of Table 2 and also include these missing references. Finally, as a minor suggestion, for the full comparison in Table 4 (Appendix C), I would recommend to include important ST references that established new SoTA at the time of their publications: ESPNet-ST (Inaguma et al. 2020), Fairseq-S2T (Wang et al.), Dual decoder Transformer (Le et al. 2020), BiKD (Inaguma et al. 2021), Adapters (Le et al., 2021), TDA (Du et al., 2022), just a few off the top of my head.

**Reproducibility:**

3: Could reproduce the results with some difficulty. The settings of parameters are underspecified or subjectively determined; the training/evaluation data are not widely available.

**Reviewer Confidence:**

4: Quite sure. I tried to check the important points carefully. It's unlikely, though conceivable, that I missed something that should affect my ratings.

**Typos Grammar Style And Presentation Improvements:**

- L143: D_st is not defined previously
- Table 1: would be more straightforward to use "Weighted shrinking" instead of "Blank label" as the former term is already used and clearly described in previous text

---

> ### Author Rebuttal · Authors · 2023-08-29
>
> Thanks for your constructed reviews! We truly appreciate the time and effort you invested in understanding our work! Below are our responses to your concerns and questions:
>
> Q1: This paper misses important references to highly related work and thus fails to properly position itself with respect to the existing literature.
>
> A1: We appreciate your suggestions regarding the inclusion of important references. However, we believe there might be some misunderstandings concerning our method and some of the referenced works. To clarify, our paper is focused on the specific topic of **shrinking speech features** to address the length gap. Given the paper's short length (only 4 pages), we feel it is neither appropriate nor practical to broaden the scope of related references and to discuss all CTC-related methods. We believe we have included all highly relevant works in our related work discussion and in our comparisons. Concerning the references you mentioned, we would like to make the following arguments:
>
> 1. The work by Zhang et al., 2023 ("Efficient CTC Regularization via Coarse Labels for End-to-End Speech Translation") should not be considered closely related to our method. According to our understanding, this work focuses on reducing the output vocabulary size of CTC modules to improve training efficiency and does not address shrinking speech representation lengths. Therefore, it should not be categorized as "highly related work."
>
> 2. We acknowledge that "shrinking speech features" is not a concept limited to CIF-based or CTC-based approaches. However, these two approaches serve as representative methods, and comparing our work to them offers meaningful insights into the effectiveness of our technique, especially given that our method aims to address the shortcomings of the CTC-based approach. Additionally, the compressed phoneme-based method mentioned (Salesky et al., 2019, "Exploring Phoneme-Level Speech Representations for End-to-End Speech Translation") also employs a CTC-based method for compressing speech features.
>
> 3. Regarding phoneme-as-MT-input methods like Tang et al., 2021 ("Improving Speech Translation by Understanding and Learning from the Auxiliary Text Translation Task"), these are not shrinking-based methods; they simply leverage phonemes as MT inputs for multi-task learning.
>
> 4. Adaptive feature selection (Zhang et al., 2020) is indeed a method that focuses on feature shrinking, but their experiments are conducted on encoders that lack any convolutional downsampling, making a fair comparison with our method challenging.
>
> Nevertheless, we will expand our discussion of related work upon acceptance, taking advantage of the one additional page allowed.
>
> Q2: Which method adopts Conv-Transformer and which uses Conformer? Do all methods use the same model architecture in each experimental setting?
>
> A2: I apologize for the lack of clarity. In all the experimental settings except for those shown in Table 2 (and its expanded version in Table 4), we used Conv-Transformer as the acoustic encoder. Additionally, the same pre-trained modules were used across these settings.
>
> Q3: Did you perform KD with a frozen pre-trained MT model or did you add an additional MT loss and perform KD with the MT model being updated during training?
>
> A3: Both methods are used and they have different performance in different language settings. Please refer to Table 6 of Appendix C for more details on choices among different language settings (for results "W/O MTL", it means performing KD with a frozen pre-trained MT model; while for results "With MTL", it means performing KD with the MT model being updated during training).
>
> Q4: The comparison to SoTA methods has some issues. ...in reality Table 2 only contains "modality alignment" methods.
>
> A4: Sorry for the confusion caused. We will revise the description in revision. And thanks for introducing two concurrent works (Le et al. 2023 and Ouyang et al. 2023) and more other important ST references, we will include their results in revision.
>
> Q5: Is previous work comparable with this work in terms of ASR and MT pre-training? Are all methods in Table 1 pre-trained in the same way?
>
> A5: In Table 1, the answer is yes, as we re-implemented all the compared methods in a fair manner, where the only difference among them is how they shrink the speech features. As for the results in Table 2, since most of the listed results differ in the used architectures/datasets, it is difficult to maintain an exact fair comparison and we have tried our best by using similar model size and data amount.
>
> Q6: Why is the choice of vocabulary size 16000?
>
> A6: It is an empirical setup following our previous preliminary experiments.
>
> Q7: Table 1: What are the model architectures used in these methods? Why used Diff<2 instead of the percentage of lengths between shrunk speech and text?
>
> A7: All the methods listed in Table 1 employ a Conv-Transformer architecture for the acoustic encoder. We chose to use the "Diff<2" metric as opposed to percentage-based metrics for lengths between shrunk speech and text, following the precedent set in previous works, such as "Bridging the Modality Gap for Speech-to-Text Translation" by Liu et al. (2020) and "RealTranS: End-to-End Simultaneous Speech Translation with Convolutional Weighted-Shrinking Transformer" by Zeng et al. (2021). This metric effectively indicates the number of samples that have been successfully aligned, offering a more intuitive basis for comparison.
>
> Q8: Do you have an intuition why weighted shrinking has a much bigger impact on En-Fr while it has only a marginal effect on En-De?
>
> A8: First of all, we believe that drawing a definitive conclusion at this point may be premature, as we have only conducted a limited comparison between MUSTC En-Fr and En-De. The observed difference in performance could be attributed to several factors, such as data distribution or model architecture. If the conclusion holds true after further validation, we would consider potential explanations such as language similarity or issues related to word reordering.
>
> Q9: What are the differences in AdaTranS between Table 1 and Table 2 that leads to the gain of 2 BLEU points in Table 2?
>
> A9: The differences include: 1) replacing Conv-Transformer with Conformer as acoustic encoder; 2) addition of cross-modal contrastive learning and KD.
>
> Q10: The label <BD> corresponds to one or multiple consecutive frames? What is the shrinking method in the latter case?
>
> A10: It corresponds to one frame. Every time we meet a frame with <BD> label, we should shrink the speech features. In other words, the number of <BD> label predicted should be the final length of the shrunk features.
>
> Q11: Did you try using simpler shrinking operations such as averaging instead of weighted shrinking after the boundary predictor module?
>
> A11: The results "- Blank Label" in Table 1 are achieved with averaging, as stated in Line 256-258.

---

### Official Review · Reviewer_T3ym · 2023-08-11

**Soundness:** 3

**Ethical Concerns:**

Yes

**Excitement:**

3: Ambivalent: It has merits (e.g., it reports state-of-the-art results, the idea is nice), but there are key weaknesses (e.g., it describes incremental work), and it can significantly benefit from another round of revision. However, I won't object to accepting it if my co-reviewers champion it.

**Paper Topic And Main Contributions:**

This article proposes an improved shrinking method based on Connectionist Temporal Classification (CTC) to address the length gap between speech and text. Experimental results demonstrate that this approach outperforms other methods in terms of both performance and speed. However, the innovation in this work is somewhat lacking, and there is a deficiency in explaining certain experimental phenomena. For instance, why is it better for the lengths of speech and text representations to be closer? Is it because shrinked speech representation is more similar to text embedding? Additionally, does bridging the length gap also narrow the representation space gap?

**Questions For The Authors:**

1) Can boundary predictor loss make CTC predictor better?
2) What about training with only boundary predictor loss without CTC loss? I mean you can get the word boundary from a pretrained CTC model.

**Reasons To Accept:**

This article proposes an improved shrinking method based on Connectionist Temporal Classification (CTC) to address the length gap between speech and text. Experimental results demonstrate that this approach outperforms other methods in terms of both performance and speed.

**Reasons To Reject:**

The innovation in this work is somewhat lacking, and there is a deficiency in explaining certain experimental phenomena. For instance, why is it better for the lengths of speech and text representations to be closer? Is it because shrinked speech representation is more similar to text embedding? Additionally, does bridging the length gap also narrow the representation space gap?

**Reproducibility:**

4: Could mostly reproduce the results, but there may be some variation because of sample variance or minor variations in their interpretation of the protocol or method.

**Reviewer Confidence:**

4: Quite sure. I tried to check the important points carefully. It's unlikely, though conceivable, that I missed something that should affect my ratings.

---

> ### Author Rebuttal · Authors · 2023-08-29
>
> Thanks for your constructed reviews! We truly appreciate the time and effort you invested in understanding our work! Below are our responses to your questions:
>
> Q1: Why is it better for the lengths of speech and text representations to be closer? Is it because shrinked speech representation is more similar to text embedding?
>
> A1: We discussed this between lines 46 and 58. To be explicit, under the paradigm of "pre-training on ASR and MT data, then fine-tuning on ST", making the lengths of speech representations similar to those of text embeddings can facilitate the transfer of pre-trained knowledge from MT models.
>
> Q2: Does bridging the length gap also narrow the representation space gap?
>
> A2: According to our experiment shown in Figure 4 of Appendix B, shrinking does indeed result in better source-target alignment, which helps to narrow the representation space gap. On the other hand, many other methods also aim to bridge this gap. Our experiment in Table 2 demonstrates that our method is complementary to those approaches.
>
> Q3: Can boundary predictor loss make CTC predictor better?
>
> A3: As both the prediction loss and the CTC loss are jointly trained using a shared acoustic encoder, the learning derived from the boundary prediction loss should influence the CTC predictor. However, we believe this effect is likely to be minor, given that the boundary prediction loss is entirely guided by the CTC loss. Further experimentation may be needed to definitively determine whether this effect is positive or not.
>
> Q4: What about training with only boundary predictor loss without CTC loss? I mean you can get the word boundary from a pretrained CTC model.
>
> A4: We chose to jointly train using both the CTC loss and the boundary predictor loss for two main reasons:
> 1) The CTC loss serves as a form of regularization. Retaining this loss helps the acoustic encoder concentrate on modeling speech effectively, thereby benefiting downstream tasks.
> 2) Since both loss functions share the same input modules, i.e. the acoustic encoder, joint training is a more efficient approach.

---

### Meta-Review · Area_Chair_6UZ3 · 2023-09-15

**Recommendation:** 4

**Metareview:**

Paper addresses the modality gap between speech and text representations when using pre-trained models for speech2text translation. It explores the use of boundary predictions (from CTC) in the shrinking process to reduce gap between speech sequence length and text sequence length. The paper is well-written and method proposed (as well experiments presented) are overall convincing. However the positioning related to previous approaches (to address modality gap for S2T translation) could be improved and this makes evaluation of the originality of the work difficult.
Moreover, questions were raised about the accuracy of the CTC boundary prediction and the use of pre-trained forced aligners instead. Finally, the way additional alignment objectives are included is unclear and some improvement on this part would be needed.

---

### Decision · Program_Chairs · 2023-10-07

**Decision:**

Accept-Findings

**Comment:**

Paper addresses the modality gap between speech and text representations when using pre-trained models for speech2text translation. It explores the use of boundary predictions (from CTC) in the shrinking process to reduce gap between speech sequence length and text sequence length. The paper is well-written and method proposed (as well experiments presented) are overall convincing. However the positioning related to previous approaches (to address modality gap for S2T translation) could be improved and this makes evaluation of the originality of the work difficult.
Moreover, questions were raised about the accuracy of the CTC boundary prediction and the use of pre-trained forced aligners instead. Finally, the way additional alignment objectives are included is unclear and some improvement on this part would be needed.